# Epidemiology, Clinical Characteristics, Diagnostic Work Up, and Treatment Options of *Leishmania* Infection in Kidney Transplant Recipients: A Systematic Review

**DOI:** 10.3390/tropicalmed7100258

**Published:** 2022-09-22

**Authors:** Evaldo Favi, Giuliano Santolamazza, Francesco Botticelli, Carlo Alfieri, Serena Delbue, Roberto Cacciola, Andrea Guarneri, Mariano Ferraresso

**Affiliations:** 1Kidney Transplantation, Fondazione IRCCS Ca’ Granda Ospedale Maggiore Policlinico, 20122 Milan, Italy; 2Department of Clinical Sciences and Community Health, Università degli Studi di Milano, 20122 Milan, Italy; 3Nephrology, Dialysis and Transplantation, Fondazione IRCCS Ca’ Granda Ospedale Maggiore Policlinico, 20122 Milan, Italy; 4Department of Biomedical, Surgical and Dental Sciences, Università degli Studi di Milano, 20122 Milan, Italy; 5Surgery, King Salman Armed Forces Hospital, Tabuk 47512, Kingdom of Saudi Arabia; 6HPB Surgery and Transplantation, Fondazione PTV, 00133 Rome, Italy

**Keywords:** *Leishmania*, Leishmaniasis, kidney transplant, allograft, infection, treatment, tropical disease, complications, outcomes

## Abstract

Current knowledge on *Leishmania* infection after kidney transplantation (KT) is limited. In order to offer a comprehensive guide for the management of post-transplant Leishmaniasis, we performed a systematic review following the latest PRISMA Checklist and using PubMed, Scopus, and Embase as databases. No time restrictions were applied, including all English-edited articles on Leishmaniasis in KT recipients. Selected items were assessed for methodological quality using a modified Newcastle–Ottawa Scale. Given the nature and quality of the studies (case reports and retrospective uncontrolled case series), data could not be meta-analyzed. A descriptive summary was therefore provided. Eventually, we selected 70 studies, describing a total of 159 cases of Leishmaniasis. Most of the patients were adult, male, and Caucasian. Furthermore, they were frequently living or travelling to endemic regions. The onset of the disease was variable, but more often in the late transplant course. The clinical features were basically similar to those reported in the general population. However, a generalized delay in diagnosis and treatment could be detected. Bone marrow aspiration was the preferred diagnostic modality. The main treatment options included pentavalent antimonial and liposomal amphotericin B, both showing mixed results. Overall, the outcomes appeared as concerning, with several patients dying or losing their transplant.

## 1. Introduction

Leishmaniasis represents one of the most neglected infectious diseases worldwide, largely affecting individuals with disadvantaged social backgrounds residing in less economically developed countries, often afflicted by malnutrition, poor residency conditions, and generalized lack of health care resources [1,2]. The term Leishmaniasis encompasses a group of parasite-associated diseases with cutaneous, mucocutaneous, or visceral manifestations, primarily caused by obligate intracellular protozoa of the genus *Leishmania* [3,4]. Although most cases remain outside formal registries, Leishmaniasis is currently endemic in more than 80 countries, particularly in the tropics, subtropics, and southern regions of Europe, with 350 million people at risk and 1.6 million new infections every year [2,3,4,5,6]. Given the ongoing global climate and environmental changes, it is likely that the geographic range of the vectors of *Leishmania* and the areas in the world where *Leishmania* can be found will further expand. Leishmaniasis is usually acquired through the bite of a female *Phlebotomus* or *Lutzomyia* sandfly, introducing the promastigotes of *Leishmania* into the bloodstream [7]. Being obligate intracellular pathogens, promastigotes are carried by circulating monocytes, in which they reproduce as amastigotes. Inside the cell, the antigens produced by the parasites inhibit the activation of the inducible nitric oxide synthase as much as several cytokine-primed cellular functions involved in the killing of infectious agents [8]. Eventually, the macrophage ruptures and releases dozens of amastigotes, which, in turn, can infect both circulating and fixed macrophages in different tissues [9]. It is the latter event that defines the primary site of the disease and the clinical subtype of Leishmaniasis. Infected circulating monocytes may be sucked by other vectors, starting reproduction as promastigotes and, therefore, completing the life cycle of the parasite.

Albeit relatively rare, post-transplant Leishmaniasis is a dreadful complication, possibly leading to allograft failure and recipient death [10]. Mostly described in a kidney transplant (KT) setting (up to 77% of the cases), *Leishmania* infection has also been reported after liver, heart, lung, pancreas, and bone marrow transplantation [7,8,9]. In the Mediterranean basin, the highest prevalence of Leishmaniasis among solid organ transplant recipients is recorded in Italy, Spain, and France. Such an observation is not, in any case, surprising, as all these countries have recently witnessed massive migrations from North Africa or the Middle East, have high-volume transplant centers, and operate in endemic areas [3,7]. As in the general population, all forms of Leishmaniasis have been described in transplant recipients, including visceral (VL), mucocutaneous (MCL), and cutaneous (CL) Leishmaniasis [3]. The increased susceptibility to *Leishmania* infection and the higher risk of severe complications observed among transplanted patients can be ascribed to the chronic effects of anti-rejection prophylaxis. Indeed, post-transplant immunosuppression inhibits both natural and acquired immunity, thus reducing the defense mechanisms against intracellular microorganisms, especially those associated with the Th1-driven antigen-specific immune response that generally favor amastigotes’ eradication in non-immunocompromised hosts [3]. The administration of the calcineurin inhibitors (CNI), cyclosporin and tacrolimus, is associated with a significant inhibition of antigen-specific CD4+ T-cell response and a generalized downregulation of several cytokines involved in cell-mediated immunity, such as IL-2, IL-4, TNF-α, and IFN-γ. The antiproliferative agent, mycophenolate mofetil (MMF), determines direct inhibition of T- and B-cell proliferation in response to antigen stimulation through the blockage of de novo purine synthesis. Azathioprine (AZA) inhibits central promyelocytes proliferation, thus decreasing the number of circulating monocytes. The mammalian target of rapamycin inhibitors (mTORi) everolimus and sirolimus act by blocking the cytokine-dependent phase of the cell-division cycle (namely, from G1 to S) and downregulating the secretion of IL-2. Corticosteroids inhibit T cells’ and antigen-presenting cells’ cytokines release and reduce the expression of several cytokine receptors located on the surface of target cells, including IL-1, IL-2, IL-3, IL-6, TNF-α, and IFN-γ [11]. In transplant recipients, the development of symptomatic Leishmaniasis recognizes three possible pathogenetic mechanisms [1]. In the first scenario, *Leishmania* infection is acquired after transplantation, determining the rapid onset of a clinically overt disease. This situation mostly occurs in patients living in endemic countries or in individuals with recent travel to endemic regions. Alternatively, the disease can be caused by a dormant *Leishmania* infection acquired before transplantation and reactivated under immunosuppression. Although extremely rare, the third option is the acquisition of *Leishmania* directly from the donor through the transplanted organ or through the administration of infected blood products [1,7,12]. Considering the recent changes in migration routes, travel habits, and transplant activities worldwide, such a route of transmission will certainly become more relevant in the future, possibly requiring reconsideration of current donor screening protocols.

Clinical features of VL, the most severe form of the disease, include fever, hepatomegaly, splenomegaly, and pancytopenia [3]. Overall, symptoms and signs of *Leishmania* infection are similar in both transplanted and non-transplanted patients. Nevertheless, due to the rarity of the disease, the limited knowledge among non-specialized clinicians (especially in non-endemic regions), the increased susceptibility to viral infections with overlapping manifestations, and the frequent occurrence of drug-related myelotoxicity, immunosuppressed subjects often experience a delay in diagnosis and treatment. Bone marrow aspiration (BMA), usually carried out to investigate possible causes of pancytopenia, represents the cornerstone of the diagnostic work up. A polymerized chain reaction (PCR) can also be used for diagnosis as much as for the definition of *Leishmania* species. Serological tests for antigens or antibodies detection are generally performed to screen individuals at risk or to confirm donor-derived infections [13]. For many years, pentavalent antimonial has been widely adopted as a first-line treatment of VL. Given the high incidence of adverse events (up to 34% in some series), it is being replaced by liposomal amphotericin B, which seems to be more effective and better tolerated [7].

To date, no meta-analyses or systematic reviews on *Leishmania* infection after KT have been published. Therefore, we aimed to comprehensively review available literature on epidemiology, clinical characteristics, diagnostic work up, and treatment options of Leishmaniasis in KT recipients.

## 2. Materials and Methods

We conducted a systematic review according to the latest Preferred Reporting Items for Systematic Reviews and Meta-Analyses (PRISMA) Checklist. PubMed, Embase, and Scopus were searched in March 2022 for any papers (including congress abstracts) reporting on patients with Leishmaniasis after KT. No time limits were applied. The following keyword combinations were used: “Leishmania AND kidney transplant”, “Leishmania AND renal transplant”, “Leishmaniasis AND kidney transplant”, or “Leishmaniasis AND renal transplant”. Only manuscripts edited in English were considered.

Two different groups of authors performed the primary (EF and GS) and secondary (FB and CA) searches. Disagreements between the two groups were resolved by discussion with a third author (AG) and the senior author (MF). Duplicates and non-English articles were removed. The remainder were screened out by reading the titles and abstracts. All items potentially describing cases of patients developing Leishmaniasis after KT were assessed in full text whilst items reporting on patients with pre-transplant diagnosis of *Leishmania* infection were excluded. Only original contributions reporting on Leishmaniasis in KT recipients were considered. An additional search of reference lists was performed by SD and RC. If available, the following data were collected and transferred to a dedicated database: recipient country of origin and travel activity to endemic regions, patient ethnicity, sex, and age, donor type, immunosuppression, time from transplant to Leishmaniasis onset, time from symptoms onset to final diagnosis, *Leishmania* species, symptoms, diagnostic work up, treatment, outcomes, Leishmaniasis-specific survival, graft survival, and irreversible graft disfunction. Extracted data were transferred to a dedicated anonymized database for analysis purposes. 

Selected studies were assessed for methodological quality using a tool based on a modification of the Newcastle–Ottawa Scale as proposed by Murad et al. [14]. As suggested by the authors, questions 5 and 6 of the original questionnaire were not considered, since they were mostly relevant to cases of drug-related adverse events. Rather than using an aggregate score, we made an overall judgement considering the questions deemed most critical in the specific clinical scenario. Accordingly, the quality of the studies was classified as low, average, or high, depending on their scoring in the questionnaire: respectively, 0–2, 3–4, or 5–6 points out of a total of 6 points. 

Our systematic review considered a large majority of single case reports and some small retrospective case series. No meta-analysis could be performed as the small case series are composed of heterogeneous patients, making any summary measures meaningless. To compactly describe the literature, we reported the number for the categorical variables and the range for the continuous ones. The tables must also be considered as a compact way of describing the results from the literature. No inferences can be drawn from this study. Furthermore, as a potential limitation of the present work, we recognize the possibility that some cases of Leishmaniasis in a KT setting may have been omitted since they could have been included in papers or congress abstracts referring to solid organ transplant recipients in general. The statistical methods were assessed by an expert in biomedical statists (CA). The review was not registered. 

## 3. Results

### 3.1. Included Studies

A flow diagram summarizing included articles and selection processes is depicted in Figure 1. 

The number of reports preliminarily retrieved using each of the keyword combinations previously mentioned was 954. In more detail: Leishmania AND renal transplant, 149 (58 from PubMed, 47 from Scopus, and 44 from Embase); Leishmania AND kidney transplant, 161 (53 from PubMed, 37 from Scopus, and 71 from Embase); Leishmaniasis AND renal transplantation, 318 (102 from PubMed, 102 from Scopus, and 114 from Embase); Leishmaniasis AND kidney transplantation, 326 (97 from PubMed, 151 from Scopus, and 78 from Embase). After duplicate (*n* = 755) and non-English articles (*n* = 8) were removed, a pool of 191 items remained for further evaluation. Following the inclusion criteria previously described and after reviewing papers by title and abstract, 83 articles were identified. Studies not reporting original cases of Leishmaniasis after KT were excluded (*n* = 13). No additional reports were found through searches of references. Eventually, 70 papers were selected. No randomized clinical trials, prospective controlled studies, or prospective uncontrolled studies were identified. At the end of the process, we included 66 retrospective case reports and 4 retrospective case series. According to the modified Newcastle–Ottawa Scale, 11 items were classified as low-quality [7,15,16,17,18,19,20,21,22,23,24], 34 as average-quality [25,26,27,28,29,30,31,32,33,34,35,36,37,38,39,40,41,42,43,44,45,46,47,48,49,50,51,52,53,54,55,56,57,58], and 25 as high-quality studies [4,10,59,60,61,62,63,64,65,66,67,68,69,70,71,72,73,74,75,76,77,78,79,80,81]. In total, our analysis includes 159 cases of *Leishmania* infection after KT. The main characteristics and qualitative evaluations of the studies meeting the criteria for the systematic review are described in Table 1.

### 3.2. Epidemiology

No articles reported on the total number of KT performed over the same period in which *Leishmania* infections were diagnosed and treated. Consequently, no estimate of cumulative incidence or prevalence could be calculated.

Information regarding the country of origin or travel activity was available for 153/159 (96.2%) patients. The vast majority (*n* = 143) of KT recipients with Leishmaniasis came from or had travelled to endemic regions, especially South America, North Africa, the Middle East, India, or the Mediterranean basin [4,16,17,18,22,23,24,25,26,27,28,30,31,34,35,36,38,40,41,42,44,45,48,50,51,52,54,55,56,57,59,60,61,62,63,64,65,66,67,69,70,71,72,73,75,76,78,79,81]. Only a few subjects (*n* = 10) were from non-endemic areas [7,10,19,49,80], whilst data were not available for the others (*n* = 6) [10,20,21,43,46,47]. 

Most episodes of *Leishmania* infection were registered in Brazil (*n* = 66) [22,29,30,40,58,68,74,79]. Several cases were also reported in the Mediterranean basin (*n* = 61), such as Spain (*n* = 20) [18,27,36,41,48,53,55,61,63,68,78], Italy (*n* = 14) [31,33,35,42,44,45,51,72,81], France (*n* = 10) [25,39,64], Tunisia (*n* = 8) [7,17,34,52,62,69], Greece (*n* = 4) [54], Turkey (*n* = 3) [59,67,77], Malta (*n* = 1) [38], and Algeria (*n* = 1) [23]. Overall, European countries outside the Mediterranean basin recorded a small number of infections: Portugal (*n* = 2) [24,66], Switzerland (*n* = 1) [73], Finland (*n* = 1) [60], United Kingdom (*n* = 1) [19], and Belgium (*n* = 1) [80]. Unexpectedly, a few cases were also described by authors living in countries in which *Leishmania* is considered as endemic, including India (*n* = 5) [26,49,57,71,76], Sudan (*n* = 1) [37], Nepal (*n* = 1) [4], Kingdom of Saudi Arabia (*n* = 2) [65,70], Iran (*n* = 1) [50], and Colombia (*n* = 1) [28]. Finally, Australia contributed with a single report [56].

### 3.3. Patients’ Characteristics

Data regarding ethnicity were available for 81/159 (50.9%) recipients [4,7,10,19,22,25,26,28,29,30,33,35,37,38,39,42,43,45,52,58,61,66,68,70,73,75,80]. The most represented heritages were Caucasian (*n* = 45) [10,19,22,25,30,33,35,37,38,39,42,43,45,52,61,66,68,70,73,75,80] and Afro-Caribbean (*n* = 30) [7,10,22,30]. Six patients were Asian or Hispanic [4,22,26,28,29,58].

Sex and age at diagnosis were recorded for 157/159 (98.7%) and 156/159 (98.1%) subjects, respectively. Eventually, we were able to identify 34 female and 123 male KT recipients with *Leishmania* infection. 

Patients’ age ranged from 12 to 76 years, with a single pediatric case [23] and 18 recipients older than 60 years [10,17,31,33,35,47,58,63,64,66,70,72,73,75,81]. In the study by Alves da Silva et al. [22] enrolling 20 patients, the mean age at diagnosis was 37 (±10.7) years. Similarly, in a series of 50 subjects, de Silva et al. reported a mean age of 40 years [30]. 

The donor type was available for 112/159 (70.4%) patients: 64 received a deceased donor kidney [7,17,22,27,28,30,31,35,39,40,46,47,50,51,52,53,54,60,61,63,65,66,69,72,73,78,79,81] and 50 had a living donor [4,17,21,22,23,26,29,30,34,38,40,54,57,59,62,67,68,70,71,72,76,77,79]. 

Maintenance immunosuppression was very heterogeneous. Most patients were on a CNI-based triple-agent scheme with tacrolimus (*n* = 54) [17,20,21,23,24,25,26,27,30,31,34,40,41,49,58,59,62,66,72,73,79,81] or cyclosporin (*n* = 45) [4,17,29,30,33,35,36,39,40,45,46,47,50,51,53,54,60,61,63,64,65,69,70,71,74,75,78,79], AZA (*n* = 51) [4,18,19,20,29,30,35,37,38,42,44,46,48,51,53,54,55,56,57,60,61,62,63,64,65,67,68,69,70,71,76,78,79,80] or MMF (*n* = 66) [17,21,23,25,26,27,28,30,31,34,40,41,43,45,49,50,58,59,66,72,73,75,77,79,81], and steroids such as prednisone (*n* = 82) [18,19,21,23,25,27,29,30,31,34,36,37,39,40,41,45,46,48,49,51,53,55,56,58,61,62,63,65,68,69,70,73,74,75,78,79,80,81], prednisolone (*n* = 20) [4,24,38,42,43,44,50,57,59,64,66,67,71,72,76,77], or methylprednisolone (*n* = 7) [33,35,44,54,60]. Administration of mTORi was reported in four cases [43,72,77]. Data were missing for 14 patients [7,10,15,16,22,32,52,54] and partially reported for 8 recipients [17,26,28,47,64]. Characteristics of KT recipients with Leishmaniasis are summarized in Table 2.

### 3.4. Clinical Presentation and Laboratory Findings

The most frequently reported form of Leishmaniasis was VL (*n* = 149) [4,7,10,15,16,17,18,19,20,22,23,24,25,26,27,28,29,30,31,35,36,38,39,40,41,42,43,44,46,47,48,49,51,52,53,54,55,56,57,59,60,61,63,64,65,66,67,69,70,71,72,73,74,75,76,77,78,79,80]. Among these patients, 50 also presented with *Leishmania*-associated skin and/or mucosal lesions [4,22,23,29,30,43,49,66,74]. Isolated cases of CL or MCL were described in six [21,32,34,50,62,68] and four [33,37,45,58] recipients, respectively. 

Overall, the time between transplant and Leishmaniasis onset ranged from a few days to 246 months. 

Information regarding the time required for a definitive diagnosis was seldom reported (*n* = 35), with a time lapse ranging from 6 to 360 days [4,26,28,35,38,39,42,47,51,53,54,55,57,60,61,62,63,65,67,70,71,76,77,78,79,80,81]. 

Fever, hepatomegaly, splenomegaly, and fatigue were the most represented symptoms, being reported in 123 [4,7,10,15,17,18,22,23,25,27,28,29,30,31,35,36,38,39,40,42,43,46,47,48,49,51,52,53,54,55,56,57,59,60,61,63,64,65,66,67,69,70,71,72,73,74,75,76,77,78,79,80,81], 107 [4,7,10,18,22,23,24,26,29,30,31,35,36,39,40,42,43,46,47,48,49,51,52,53,54,54,57,59,61,63,64,65,67,71,72,73,74,75,76,77,78,79], and 87 [4,7,17,22,25,26,27,28,29,30,36,38,40,42,43,48,49,51,53,54,56,57,60,61,63,65,66,67,69,71,73,74,75,76,77,78,81] patients, respectively. Respiratory symptoms (*n* = 9) [28,42,59,60,61,76,77,78,81] or lymphadenopathy (*n* = 4) [4,27,71,79] were rarely described. 

The peripheral blood cell count was abnormal in most cases. In a series of 30 patients, Silva et al. [30] reported a mean hematocrit of 28.7%, a mean leucocyte count of 3000 × 10^9^/L, and a mean platelet count of 110 × 10^9^/L (for analysis purposes, these recipients were all considered as having pancytopenia). Pancytopenia was also described as the main laboratory finding in other 65 subjects [4,7,10,15,18,23,24,25,26,27,28,31,35,36,38,40,42,43,46,47,49,51,52,53,54,55,56,57,59,61,63,64,66,69,70,72,73,77,78,79,80,81]. Isolated anemia was reported in seven cases [17,40,64,75], isolated leukopenia in two [48,71], concomitant anemia and leukopenia in six [29,60,65,67,74,76], and associated leukopenia and thrombocytopenia in five [34,39,64,78]. Eleven cases were simply described as VL, with no data on symptoms or laboratory findings [16,19,20,41,44]. The main symptoms and laboratory findings of post-transplant Leishmaniasis are summarized in Table 3. 

### 3.5. Diagnostic Work Up

A diagnostic work up was reported in 156/159 (98%) cases. BMA was the preferred and most reliable diagnostic modality, being used in 123 patients [4,7,15,16,17,18,22,28,29,30,31,35,36,39,40,41,42,43,44,46,47,48,51,53,54,55,57,59,60,61,63,64,65,71,73,74,75,76,77,78,79,81]. Overall, the test was able to detect *Leishmania* amastigotes in 115 recipients, with 8 episodes of missed diagnosis [23,26,27,33,40,45,49,78]. For *Leishmania* typing, the use of PCR-based methods was described in 21 reports [17,23,24,25,27,28,29,30,31,32,43,54,64,66,68,73,74], whereas the presence of *Leishmania* antigens or *Leishmania*-specific antibodies was assessed by immunofluorescence or ELISA tests in 36 [7,17,18,25,28,31,32,35,37,39,45,46,47,48,51,54,55,63,64,66,73,75,78] and 26 patients [22,23,26,30,32,42,55,66,67,72,79], respectively. Additional investigations were skin or mucosal biopsy (*n* = 16) [4,21,23,33,34,37,43,45,49,50,58,62,64,66,68,74], spleen biopsy (*n* = 3) [22,30,40], gastrointestinal biopsy (*n* = 2) [24,26], lymph node biopsy (*n* = 1) [27], *Leishmania* antigens search in urinary samples (*n* = 4) [41], aqueous humor culturing (*n* = 1) [29], and bronchoalveolar lavage (*n* = 1) [60]. 

The *Leishmania* species was reported in 45 cases. Available data show that *Leishmania donovani* (*n* = 20) [4,26,27,33,37,38,39,42,44,47,48,53,56,57,63,67,71,76,80,81] and *Leishmania infantum* (*n* = 19) [7,17,18,19,24,25,43,64,66,73] were the most frequently identified parasites, followed by *Leishmania braziliensis* (*n* = 5) [29,32,58,68,74] and *Leishmania mexicana* (*n* = 1) [28]. 

### 3.6. Patient- and Transplant-Related Outcomes

Of the 159 KT recipients with *Leishmania* infection described in the literature, survival was reported for 157/159 (98.7%) [4,7,10,15,16,17,18,19,21,22,23,24,25,26,27,28,29,30,31,32,33,34,35,36,37,38,39,40,41,42,43,44,45,46,47,48,49,50,51,52,53,54,55,56,57,58,59,60,61,62,63,64,65,66,67,68,69,70,71,72,73,74,75,76,77,78,79,80,81]. Precisely, 26 deaths were recorded: 21 were directly related to Leishmaniasis [16,17,22,30,55,56,64,73,74,80], 1 was due to a cerebrovascular accident [81], and 4 remained undetermined [29,43,44,58]. Recipients’ age at the time of the exitus ranged from 32 to 73 years. Most subjects who eventually died of Leishmaniasis showed signs of systemic disease (VL); one patient had MCL [58]. Treatment details were available for 18 patients. According to the information collected, eight subjects had received liposomal amphotericin B [17,43,44,58,64,66,73,81], seven had been given pentavalent antimonial [17,28,29,42,74,78,80], one had been administered metronidazole [55], and two were not treated [16,56]. For eight patients, no information could be retrieved [22,30]. 

Analyzing data of surviving recipients, we detected 19 episodes of allograft loss [10,30,41,64,72,74,78] and 24 cases of irreversible allograft disfunction [10,22,30,41,55,64,65,72,73,78,81]. Patient- and transplant-related outcomes are synthetically described in Table 4.

### 3.7. Treatment Options and Treatment-Related Outcomes

Treatment was reported for 156/159 (98.1%) patients. Fist-line therapy mostly consisted of liposomal amphotericin B (*n* = 98) [4,10,17,20,22,24,25,26,27,30,31,32,40,41,43,44,49,54,58,59,64,66,72,73,77,79,81] or pentavalent antimonial (*n* = 44) [7,15,16,17,18,19,28,29,33,36,38,39,42,45,46,47,48,50,51,52,53,54,57,60,61,62,63,64,65,67,69,70,71,74,75,76,78,79,80]. Other options included non-liposomal amphotericin B (*n* = 6) [30,35,64,79], metronidazole (*n* = 2) [34,55], and a combination of azoles and allopurinol (*n* = 2) [37,68]. A patient with CL was successfully treated using local cryotherapy [21]. In three cases, no *Leishmania*-specific therapy was administered [16,56,64]. Among these patients, only one survived [64]. As mentioned above, liposomal amphotericin B was used in 98 cases. However, in three reports, the outcomes were not described clearly [23,24,72]. Moreover, in the studies by de Silva et al. [30] and Alves da Silva et al. [22], the results of 46 patients treated with liposomal amphotericin B were mixed with those of four subjects receiving a different treatment, making it impossible to obtain meaningful information. With all the due limitations, we concluded that liposomal amphotericin B led to complete remission in 25 recipients [4,10,26,27,31,32,40,43,49,54,58,64,72,79], partial remission in 14 [10,17,27,40,41,44,77,79,81], and no response in five [17,40,41,59,66]. Four individuals developed serious adverse reactions [20,25,44,73], whereas eight patients died despite treatment [17,43,44,58,64,66,73,81]. The administration of pentavalent antimonial (*n* = 44) was associated with complete remission (*n* = 15) [16,33,39,45,48,53,57,60,64,70,76,78,79], partial remission (*n* = 9) [18,19,29,36,51,54,67,71], or treatment failure (*n* = 9) [17,28,42,50,62,64,74,78,80]. Eleven patients experienced severe drug-related adverse events [7,15,38,46,47,52,61,63,65,69,75]. Overall, seven out of 44 recipients did not survive [17,28,29,42,74,78,80]. 

Information on secondary prophylaxis was omitted in virtually all the studies included. On the contrary, data regarding relapse episodes after first-line and second-line treatments were reported for 159 and 50 KT recipients, respectively. First-line and second-line treatments with treatment-specific outcomes (including relapses) are summarized in Table 5 and Table 6.

## 4. Discussion

It is well-known that solid organ transplant recipients are more prone to opportunistic infections than the general population [13]. Over the last two decades, we have witnessed a considerable rise in the cases of Leishmaniasis among KT patients. Such a concerning phenomenon is mostly due to a wider diffusion of the parasite, as well as the exponential increase in the number of patients at risk of the disease. Undoubtedly, the massive migrations from rural to urban areas, the savage requalification processes of rural and suburban zones, and the opportunity to easily travel from and to endemic regions have greatly contributed to expanding the geographical distribution of the sandfly vectors of *Leishmania*. Nevertheless, the increasing prevalence of acquired immunodeficiency conditions observed in both less and more economically developed countries as a result of expanding HIV contagion and widespread transplant activity have played a significant role [2,3,5,6]. 

Unfortunately, the studies included in our systematic review failed to provide information regarding the incidence and prevalence of *Leishmania* infection among the populations enrolled, thus limiting the epidemiological value of the present analysis. However, considering the progressive increase in the number of transplants performed worldwide, current achievements in long-term recipient and allograft survival, the wider use of powerful immunosuppressive agents, and the regained awareness of the transplant community, it is reasonable to expect that the incidence of Leishmaniasis will rise considerably. In this regard, it is paramount to promptly develop national and international registries for the implementation of infection control strategies and formal outcomes assessment.

As expected, the vast majority of Leishmaniasis reported in the literature referred to KT recipients who were living or had recently travelled to endemic countries such as Brazil, Spain, Italy, France, or Tunisia [4,16,17,18,22,23,24,25,26,27,28,30,31,34,35,36,38,40,41,42,44,45,48,50,51,52,54,55,56,57,59,60,61,62,63,64,65,66,67,69,70,71,72,73,75,76,78,79,81]. The observation that several endemic countries with large populations and well-developed transplant programs have marginally contributed to the existing literature confirms the hypothesis that most cases of post-transplant *Leishmania* infections remain unreported (or perhaps unrecognized). Once again, institutional registries could help in gathering more reliable data for future research projects.

Details on donor ethnicity were completely omitted in all the studies included in the review. On the contrary, recipient heritage was described in about half of the cases. Apparently, most of the patients developing post-transplant Leishmaniasis were Caucasian [4,7,10,19,22,25,26,28,29,30,33,35,37,38,39,42,43,45,52,58,61,66,68,70,73,75,80]. However, there is scarce evidence of any actual association between ethnicity and *Leishmania* infection susceptibility among transplanted and non-transplanted subjects. Carrasco-Antón et al. [10] reported an association between sub-Saharan African ethnicity and VL in the general population, possibly explained by genetic predisposition, but the role of genetic factors in posttransplant VL remains to be determined. More likely, the perceived disproportion in the prevalence of *Leishmania* infection among different ethnic groups reflects the fact that most reports were produced by authors residing in the Mediterranean basin or possible disparities among different minorities in their access to the KT waiting list [82].

Available data seem to suggest that *Leishmania* infection is more frequent among middle aged male KT recipients [22]. However, the reduced incidence or prevalence of the disease observed among pediatric, elderly, or female recipients may be due to differences in the numbers of KT performed in these subgroups of patients rather than actual differences in infection susceptibility [83]. Accordingly, no sex-related differences in susceptibility have been confirmed in the general population.

We found that the time between transplantation and the onset of *Leishmania* infection-related symptoms was extremely variable. Nonetheless, most patients developed the disease as a late post-transplant complication [3,62]. This observation highlights the need for a high index of suspicion during all the phases of the post-transplant follow-up, particularly in the long term, as prolonged exposure to immunosuppression may progressively increase the risk of infection. In addition, it confirms that the donor–recipient route (via the allograft) has a marginal impact on the transmission of the parasite in solid organ transplant setting.

Although it sounds reasonable to assume that the use of more powerful immunosuppressive agents increases the risk of opportunistic infections including *Leishmania* [22], the paucity of reports, the lack of details, and the heterogeneity of the anti-rejection prophylaxis protocols adopted by transplant centers worldwide make it impossible to assess the relationships between specific immunosuppressants and the risk of post-transplant Leishmaniasis [4,17,18,19,20,21,23,24,25,26,27,29,30,31,33,34,35,36,37,38,39,40,41,42,43,44,45,46,47,48,49,50,51,53,54,55,56,57,58,59,60,61,62,63,64,65,66,67,68,69,70,71,72,73,75,76,77,78,79,80,81]. Certainly, the overwhelming number of sensitized patients currently engulfing the KT waiting lists and requiring complex induction immunosuppressive schemes will represent a potential target for future infections [84]. 

In the studies included in our research, VL was the predominant form of the disease reported after KT, with only a few cases of MCL or CL [21,32,33,34,37,45,50,58,62,68]. This finding confirms that KT patients are at an increased risk of severe infectious complications compared to the general population and suggests that the combination of end-stage renal disease and drug-induced immunosuppression can significantly impair *Leishmania*-specific immune response [11,12,13,14,15,16,17,18,19,20,21,22,23,24,25,26,27,28,29,30,31,32,33,34,35,36,37,38,39,40,41,42,43,44,45,46,47,48,49,50,51,52,53,54,55,56,57,58,59,60,61,62,63,64,65,66,67,68,69,70,71,72,73,74,75,76,77,78,79,80,81,82,83,84,85]. 

As in non-transplanted patients [86], post-transplant VL presented with a wide range of subtle signs and symptoms such as fever, hepatosplenomegaly, and fatigue. Cough [28,42,59,60,61,76,77,78,81] or lymphadenopathy [4,27,71,79] may occur, but the clinical picture remains extremely vague. The lack of specific features, the indolence of the course, and the rarity and scarce knowledge of the disease (particularly in non-endemic regions) can represent a difficult challenge and cause a significant delay in both diagnosis and treatment. Indeed, our analysis shows that in most cases, more than 30 days passed before obtaining a definitive diagnosis, with a negative impact on recipient- and transplant-related outcomes [4,26,28,35,38,39,42,47,51,53,54,55,57,60,61,62,63,65,67,70,71,76,77,78,79,80,81]. 

Routine laboratory tests may undoubtedly help during the diagnostic work up, but the vast majority of patients with post-transplant *Leishmania* infection exhibit non-specific abnormalities such as mild-to-moderate pancytopenia [4,7,10,15,18,23,24,25,26,27,28,30,31,35,36,38,40,42,43,46,47,49,51,52,53,54,55,56,57,59,61,63,64,66,69,70,72,73,77,78,79,80,81] or various degrees and combinations of leukopenia [48,71], anemia [17,40,64,75], and thrombocytopenia [34,39,64,78]. Similar findings are very frequently observed among KT recipients and they are more often due to drug-related side effects (especially caused by MMF and mTORi), urinary tract infections, or reactivations of latent cytomegalovirus or Epstein-Barr virus infections [11,12,13,14,15,16,17,18,19,20,21,22,23,24,25,26,27,28,29,30,31,32,33,34,35,36,37,38,39,40,41,42,43,44,45,46,47,48,49,50,51,52,53,54,55,56,57,58,59,60,61,62,63,64,65,66,67,68,69,70,71,72,73,74,75,76,77,78,79,80,81,82,83,84,85,86,87]. Additionally, the possibility of concomitant infectious complications should always be considered. A reasonable approach would be to include *Leishmania* infection screening as a part of every second-line diagnostic protocol for fever of unknown origin (FUO) after KT. 

In line with current practice in the general population [86], our review confirms the substantial role of BMA in the diagnostic work up of FUO and *Leishmania* infection, also in KT recipients [4,7,15,16,17,18,22,28,29,30,31,35,36,39,40,41,42,43,44,46,47,48,51,53,54,55,57,59,60,61,63,64,65,71,73,74,75,76,77,78,79,81]. Indeed, the procedure demonstrated high sensitivity and specificity, promptly detecting the parasite in most cases [23,26,27,33,40,45,49,78]. Other diagnostic modalities reported in the literature were immunofluorescence-based serological tests for *Leishmania* antigens [7,17,18,25,28,31,32,35,37,39,45,46,47,48,51,54,55,63,64,66,73,75,78], ELISA for *Leishmania*-specific antibodies [22,23,26,30,32,42,55,66,67,72,79], and PCR analysis for *Leishmania* detection and typing on whole blood or tissue aspirate samples [17,23,24,25,27,28,29,30,31,32,43,54,64,66,68,73,74]. Recipients with obvious cutaneous or mucocutaneous lesions could be reliably assessed by microscopic evaluation, histology, or culture of sample tissues [4,21,23,33,34,37,43,45,49,50,58,62,64,66,68,74]. 

Unfortunately, proper *Leishmania* characterization was seldom carried out. Available data suggest a predominance of *Leishmania donovani* [4,26,27,33,37,38,39,42,44,47,48,53,56,57,63,67,71,76,80,81] and *Leishmania infantum* [7,17,18,19,24,25,43,64,66,73], with few infections sustained by *Leishmania braziliensis* [29,32,58,68,74] or *Leishmania mexicana* [28]. These findings are in line with the geographical distribution of *Leishmania*, as *Leishmania donovani* is the most represented strain in south Asia and east Africa, whereas *Leishmania infantum* is the leading strain in the Mediterranean basin, Middle East, Pakistan, Iran, and Brazil [87]. Remarkably, KT patients infected by *Leishmania braziliensis* shared a particularly aggressive course of the disease, eventually leading to death in a significant portion of subjects [29,58,74]. In the general population, infections sustained by *Leishmania braziliensis* are associated with various symptoms and signs, ranging from disfiguring cutaneous or muco-cutaneous lesions to aggressive forms of visceral disease, usually failing to respond to pentavalent antimonial [88]. Although limited, our observations support systematic characterization of *Leishmania* species in all transplant recipients with suspected or overt disease, to better define the risk of negative outcomes and personalize treatment. 

In the general population, CL and MCL represent the prevalent forms of the disease. However, in the transplanted population, a striking predominance of VL has been reported, also in individuals infected by less aggressive *Leishmania* species. Left untreated, VL is associated with a lethality rate as high as 95%. As such, systemic treatment is recommended in immunocompromised hosts. Main therapeutic agents for Leishmaniasis after KT described in the literature were pentavalent antimonial and amphotericin B [82]. Pentavalent antimonial (namely, sodium stibogluconate and meglumine antimoniate) explicate their anti-*Leishmania* action inhibiting DNA topoisomerase function, glycolytic activity, and fatty acid beta-oxidation, thus inducing metabolic imbalance and structural modifications of the parasite’s membrane. The main limitations of pentavalent antimonial use are the frequent occurrence of drug-related side effects (cardiotoxicity, bone marrow suppression, nephrotoxicity, acute pancreatitis, and abnormal liver function tests), intravenous route of administration, and increasing global resistance [89]. Amphotericin B is a polyene antifungal compound with a broad range of activity against yeasts, molds, and protozoa, including *Leishmania*. Amphotericin B binds to the ergosterol of the fungal cell membrane, leading to ion leakage and cell death [90]. Exhibiting a better safety profile, liposomal amphotericin B has progressively replaced amphotericin B deoxycholate, and it is currently used as a first-line treatment of VL in patients with impaired renal function [3]. As demonstrated in the general population, our analysis suggests that liposomal amphotericin B is more effective than pentavalent antimonial in KT recipients [3]. Another emerging option for the treatment of VL is the alkyl-lysophospholipid miltefosine [41,91]. There is still limited knowledge of the mechanism of action of miltefosine, but experimental models support the hypothesis that it may trigger and enhance programmed cell death (apoptosis) in both metazoans and protozoans. Some authors also believe that miltefosine may act by inhibiting phosphatidyl choline synthesis during the processes involved in the formation of the cell membrane [92]. The efficacy and safety of miltefosine for the treatment of VL have been primarily evaluated by Sundar and Ollario in non-transplanted patients. They showed that miltefosine administration was overall well tolerated, with the occurrence of mild gastrointestinal side effects and few episodes of vomiting, diarrhea, or acute nephrotoxicity [93]. Current experience in a KT setting remains anecdotal and not convincing [41]. Awaiting additional data, it seems reasonable to restrict the use of miltefosine to KT patients not suitable for pentavalent antimonial or amphotericin B. On the contrary, available evidence suggests wide administration of amphotericin B as a first-line treatment of post-transplant VL. Proper identification of *Leishmania* species could improve treatment-related outcomes, particularly for those patients with limited response to first-line therapy.

Despite recent advancements in diagnostics and therapeutics, the outcomes of KT recipients with Leishmaniasis remain concerning, particularly in the case of systemic disease. As a matter of fact, our analysis shows that about 25% of the patients developing *Leishmania* infection during the post-transplant course eventually died, regardless of the treatment received [29,43,44,58,81]. The occurrence of allograft loss [8,30,41,64,72,74,78] or irreversible allograft dysfunction [8,22,30,41,55,64,65,72,73,78,81] was also concerning, and further emphasizes the need for optimized diagnosis and treatment.

## 5. Conclusions

To the best of our knowledge, this is the first systematic review on *Leishmania* infection after KT. Due to the lack of properly designed studies and large populations databases, performing a meaningful meta-analysis was not possible. Nevertheless, we herein reported a comprehensive and updated reference that could serve as a basis for further research projects, hopefully guiding the clinicians involved in the care of this complex group of patients in the case of suspected or overt Leishmaniasis. The institution of formal national and international registries is vital for the optimization of both management and outcomes.

## Figures and Tables

**Figure 1 tropicalmed-07-00258-f001:**
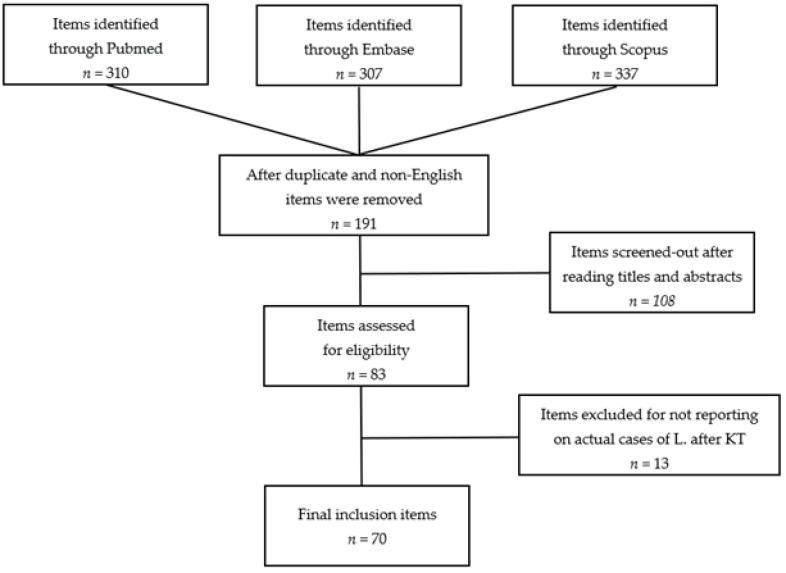
Flow diagram of the systematic review (L., Leishmaniasis; KT, kidney transplant).

**Table 1 tropicalmed-07-00258-t001:** Characteristics of studies meeting the criteria for the systematic review.

Study	Design	Period	Sample Size	Quality	Leishmaniasis
Broeckaert-Van Orshoven et al. [80]	R-CR	1979	1	H	VL
Ma et al. [56]	R-CR	1979	1	A	VL
Aguado et al. [55]	R-CR	1986	1	A	VL
Fernandez-Guerrero et al. [16]	R-CR	1987	2	L	VL
Lamas et al. [48]	R-CR	1987	1	A	VL
Donovan et al. [38]	R-CR	1990	1	A	VL
Kher et al. [76]	R-CR	1991	1	H	VL
Jokipii et al. [60]	R-CR	1992	1	H	VL
Moulin et al. [39]	R-CR	1992	1	A	VL
Halim et al. [65]	R-CR	1993	1	H	VL
Torregrosa et al. [53]	R-CR	1993	1	A	VL
Portoles et al. [78]	R-CR	1994	3	H	VL
Mittal et al. [57]	R-CR	1995	1	A	VL
Moroni et al. [42]	R-CR	1995	1	A	VL
Gòmez Campderà et al. [47]	R-CR	1996	1	A	VL
Sharma et al. [71]	R-CR	1996	1	H	VL
Torrus et al. [15]	R-CR	1996	1	L	VL
Apaydin et al. [67]	R-CR	1997	1	H	VL
Alrajhi et al. [37]	R-CR	1998	1	A	ML
Berenguer et al. [63]	R-CR	1998	1	H	VL
Boletis et al. [54]	R-CR	1998	4	A	VL
Esteban et al. [46]	R-CR	1998	1	A	VL
Roustan et al. [19]	R-CR	1998	1	L	VL
Hernàndez-Pérez et al. [18]	R-CR	1999	2	L	VL
Hueso et al. [75]	R-CR	1999	1	H	VL
Llorente et al. [61]	R-CR	2000	1	H	VL
Borgia et al. [33]	R-CR	2001	1	A	MCL
Hussein et al. [70]	R-CR	2001	1	H	VL
Gontijo et al. [74]	R-CR	2002	1	H	VL + MCL
Fernandes et al. [68]	R-CR	2002	1	H	CL
Sabbatini et al. [51]	R-CR	2002	1	A	VL
Tavora et al. [29]	R-CR	2002	1	A	VL
Valente et al. [35]	R-CR	2002	1	A	VL
Ersoy et al. [59]	R-CR	2003	1	H	VL
Basset et al. [64]	M-U-R-CS	2005	8	H	VL
Vinhal J. et al. [32]	R-CR	2007	1	A	CL
Oliveira et al. [40]	R-CR	2008	4	A	VL
Oliveira et al. [79]	R-CR	2008	8	H	VL
Dettwiler et al. [73]	R-CR	2010	1	H	VL
Harzallah et al. [69]	R-CR	2010	1	H	VL
Veroux et al. [72]	R-CR	2010	5	H	VL
Zumrutdal et al. [77]	R-CR	2010	1	H	VL
Mestra et al. [28]	R-CR	2011	1	A	VL
Orofino et al. [36]	R-CR	2011	1	A	VL
Oussalah et al. [23]	R-CR	2011	1	L	VL
Simon et al. [44]	R-CR	2011	2	A	VL
Trabelsi et al. [52]	R-CR	2011	1	A	VL
Baglieri e Scuderi et al. [45]	R-CR	2012	1	A	ML
Jha e Chugh et al. [4]	R-CR	2012	1	H	VL
Rahbar et al. [50]	R-CR	2012	1	A	CL
Alves da Silva et al. [22]	S-C-R-CS	2013	20	L	VL
Yaich et al. [62]	R-CR	2013	1	H	CL
Yucel et al. [43]	R-CR	2013	1	A	VL + CL
Bouchekoua et al. [7]	R-CR	2014	1	L	VL
Pedroso et al. [81]	R-CR	2014	1	H	VL
Tuon et al. [58]	R-CR	2014	1	A	MCL
de Silva et al. [30]	M-U-R-CS	2015	30	A	VL
Duvignaud et al. [25]	R-CR	2015	1	A	VL
Mahesh et al. [49]	R-CR	2016	1	A	VL + MCL
Carrasco-Antòn et al. [10]	R-CR	2017	6	H	VL
El Jeri et al. [17]	R-CR	2017	3	L	VL
Pérez-Jacoiste Asin et al. [41]	S-U-R-CS	2017	5	A	VL
Charfi et al. [20]	R-CR	2018	1	L	VL
Clavijo Sanchez et al. [27]	R-CR	2018	2	A	VL
Silva et al. [24]	R-CR	2019	1	L	VL
Marques et al. [66]	R-CR	2020	1	H	VL + MCL
Gembillo et al. [31]	R-CR	2021	1	A	VL
Imen et al. [34]	R-CR	2021	1	A	CL
Azzabi et al. [21]	R-CR	2022	1	L	CL
Rana et al. [26]	R-CR	2022	1	A	VL

Abbreviations: A, average-quality; CL, cutaneous Leishmaniasis; H, high-quality; L, low-quality; MCL, mucocutaneous Leishmaniasis; M-U-R-CS, multi-center uncontrolled retrospective case series; R-CR, retrospective case-report; S-C-R-CS, single-center controlled retrospective case series; S-U-R-CS, single-center uncontrolled retrospective case-series; VL, visceral Leishmaniasis.

**Table 2 tropicalmed-07-00258-t002:** Summary of the characteristics of the case reports and case series of *Leishmania* infection after kidney transplantation (summaries based on individual cases should not be considered as an estimate of the “real world”).

Variables	Range or Number of Patients (*n* = 159)
Lived in/visited endemic country	143
Sex M/F	123/34
Age	12–76
Caucasian/Afro-Caribbean/Asian or Hispanic/NA	45/30/6/78
Deceased/living donor	64/50
Tacrolimus	54
Cyclosporin	45
Prednisone	82
Prednisolone	20
Methylprednisolone	7
Mycophenolate	66
Azathioprine	51
Everolimus/Sirolimus	4

Abbreviations: F, female; M, male; NA, not available.

**Table 3 tropicalmed-07-00258-t003:** Summary of the symptoms and laboratory findings of the case reports and case series of *Leishmania* infection after kidney transplantation (summaries based on individual cases should not be considered as an estimate of the “real world”).

Variables	Patients (*n* = 159)
Skin lesions	55
Mucosal lesions	8
Fever	123
Hepatomegaly and/or splenomegaly	107
Weakness	87
Respiratory symptoms	9
Lymphadenopathies	4
Anemia	108
Thrombocytopenia	100
Leukopenia	108
Anemia and leukopenia without thrombocytopenia	6
Leukopenia and thrombocytopenia without anemia	5
Anemia alone	7
Leukopenia alone	2
Pancytopenia	95

**Table 4 tropicalmed-07-00258-t004:** Summary of the patient- and transplant-related outcomes of the case reports and case series of *Leishmania* infection after kidney transplantation (summaries based on individual cases should not be considered as an estimate of the “real world”).

Outcome	Patients (*n* = 159)
Alive	131
Dead	26
Data not available	2
Graft loss	19
Irreversible graft disfunction	24

**Table 5 tropicalmed-07-00258-t005:** Summary of first-line treatment options and treatment-related outcomes of the case reports and case series of *Leishmania* infection after kidney transplantation (summaries based on individual cases should not be considered as an estimate of the “real world”).

Treatment	Patients (*n* = 159)	Complete Remission	Partial Remission	No Response	Relapse	Treatment Duration	Drug-Related SAE	NA	Alive	Dead	NA
Liposomal amphotericin B	98	25	2	3	14	5–180 days	4	50	44	8	46
Non-liposomal amphotericin B	6	1	0	1	2	13 days	0	2	4	2	0
Pentavalent antimonial	44	15	2	7	9	5–28 days	11	0	36	7	0
Azoles +/− Allopurinol	2	1	0	0	1	28–90 days	0	0	4	0	0
Allopurinol	0	0	0	0	0	NA	0	0	0	0	0
Metronidazole	2	0	1	1	0	7–180 days	0	0	1	1	0
Other (Cryotherapy)	1	1	0	0	0	NA	0	0	1	0	0
Untreated	3	0	0	3	0	-	0	0	1	2	0
NA	3	0	0	0	0	-	0	3	2	0	1

Abbreviations: NA, not available; SAE, serious adverse event.

**Table 6 tropicalmed-07-00258-t006:** Summary of second-line treatment options and treatment-related outcomes of the case reports and case series of *Leishmania* infection after kidney transplantation (summaries based on individual cases should not be considered as an estimate of the “real world”).

Treatment	Patients (n = 159)	Complete Remission	Partial Remission	No Response	Relapse	Treatment Duration	Drug-Related SAE	Alive	Dead	NA
Liposomal amphotericin B	22	16	0	0	6	5–120	0	19	3	0
Non-liposomal amphotericin B	1	1	0	0	0	NA	0	1	0	0
Pentavalent antimonial	14	6	0	5	3	20–30	0	5	2	7
Azoles +/− Allopurinol	6	5	0	0	1	21–42	0	6	0	0
Allopurinol	2	2	0	0	0	NA	0	2	0	0
Cryotherapy	1	1	0	0	0	NA	0	1	0	0
Miltefosine	3	1	0	0	2	NA	0	3	0	0
Untreated	1	0	0	1	0	-	0	0	1	0

Abbreviations: NA, not available; SAE, serious adverse event.

## Data Availability

Not applicable.

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
