# Peer review of "Epidemiology, Clinical Characteristics, Diagnostic Work Up, and Treatment Options of *Leishmania* Infection in Kidney Transplant Recipients: A Systematic Review"

_tropicalmed, 2022, doi:10.3390/tropicalmed7100258_

Round 1
Reviewer 1 Report
Thank you for the opportunity to review this paper. I would recommend to include the limitations of this systematic review, as some cases of leishmaniasis in the transplant scenario would already be described in papers regarding solid organ transplant recipients, although not included in a specific series in KT recipients (page 3 Line 144). It should also be considered to give some information related to relapse and secondary prophylaxis (paper in attached).
Some suggestions:
o Page 2 Line 94. Although not related to KT, I would recommend adding some more comments regarding the risk of donor transmission: Please check - Donor acquired visceral leishmaniasis following liver transplantation. Dhaliwal A, Chauhan A, Aggarwal D, et al. Frontline Gastroenterology 2021;12:690–694.
o Page 3 Line 103. PCR is usually used for diagnosis but not screening, as in latent infection parasitemia can be low and intermittent
o Page 3 Line 109. There are some guidelines in the management of Leishmaniasis in transplant recipients, not specific to KT – but this wouldn’t affirm there are no clinical guidelines. Per example: Clemente WT, Mourao PHO, Lopez-Medrano F, Schwartz BS, Garcia-Donoso C, Torre-Cisneros J. Visceral and cutaneous leishmaniasis recommendations for solid organ transplant recipients and donors. Transplantation. 2018;102:S8–S15.
o Page 6 Line 200. Please add some comments in discussion regarding the lack of LV in KT reported cases in endemic countries
o Page 6 line 211. Please supply some comments in the discussion regarding why is the majority of cases described in men.
o Page 8 Line 284. Please check the writing of species Leishmania Donovani - Leishmania donovani
o Page 8 Line 270 Diagnostic workout should include comments in the discussion part - Serology per example
o Table 5 should include information regarding the treatment duration. This is important when one considers the risk of relapse

Author Response
We would like to thank Reviewer 1 for the positive feedbacks as much as for the valuable suggestions.
Q1) I would recommend to include the limitations of this systematic review, as some cases of leishmaniasis in the transplant scenario would already be described in papers regarding solid organ transplant recipients, although not included in a specific series in KT recipients.
A1) As recommended, we recognized the possibility that some studies may have been omitted (please, see Materials and methods section).
Q2) It should be considered to give some information related to relapse and secondary prophylaxis.
A2) As suggested, we included information on relapses and secondary prophylaxis (please, see Results, Table 5, and Table 6).
Q3) Although not related to KT, I would recommend adding some more comments regarding the risk of donor transmission.
A3) As suggested, we added a sentenc rearding the risk of donor transmission in the Introduction: "Although extremely rare, the third option is the acquisition of Leishmania directly from the donor through the transplanted organ or through the administration of infected blood products. Considering the recent changes in migration routes, travel habits, and transplant activities worldwide, such route of transmission will certainly become more relevant in the future, possibly requiring reconsideration of current donor screening protocols".
Q4) PCR is usually used for diagnosis but not screening, as in latent infection parasitemia can be low and intermittent.
A4) As correcty pointed out by Reviewer 1, we specified that PCR is used for diagnosis and / or typing rather than screening (please, see Introduction).
Q5) There are some guidelines in the management of Leishmaniasis in transplant recipients, not specific to KT – but this wouldn’t affirm there are no clinical guidelines.
A5) We agree with Reviewer 1. As such, we changed the sentence referring to guidelines in the Introduction: "To date, no meta-analyses or systematic reviews on Leishmania infection after KT have been published".
Q6) Please add some comments in discussion regarding the lack of LV in KT reported cases in endemic countries.
A6) Please, check the Discussion: "As expected, the vast majority of Leishmaniasis reported in the literature referred to KT recipients who were living or had recently travelled to endemic countries such as Brazil, Spain, Italy, France, or Tunisia. The observation that several endemic countries with large populations and well-developed transplant programs have marginally contributed to the existing literature, confirm the hypothesis that most cases of post-transplant Leishmania infections remain unreported (or perhaps unrecognized). Once again, institutional registries could help gathering more reliable data for next research projects".
Q7) Please supply some comments in the discussion regarding why is the majority of cases described in men.
A7) Please, consider the following paragraph in the Discussion: "Available data seem to suggest that Leishmania infection is more frequent among middle aged male KT recipients. However, the reduced incidence or prevalence of the disease observed among pediatric, elderly, or female recipients may be due to differences in the number of KT performed in these subgroups of patients rather than actual differences in infection susceptibility. Accordingly, no sex-related differences in susceptibility have been confirmed in the general population.
Q8) Please check the writing of species Leishmania Donovani - Leishmania donovani.
A8) Done.
Q9) Diagnostic workout should include comments in the discussion part.
A9) The paragraphs related to diagnostic work up in the Discussion section has been expanded: "Routine laboratory tests may undoubtedly help during the diagnostic work up, but the vast majority of patients with post-transplant Leishmania infection exhibit non-specific abnormalities such as mild-to-moderate pancytopenia or various degrees and combinations of leukopenia, anemia, and thrombocytopenia. Similar findings are very frequently observed among KT recipients and they are more often due to drug-related side effects (especially caused by MMF and mTORi), urinary tract infections, or reactivations of latent cytomegalovirus or Epstein-Barr virus infections. Also, the possibility of concomitant infectious complications should be always considered. A reasonable approach would be to include Leishmania infection screening as a part of every second-line diagnostic protocol for fever of unknown origin (FUO) after KT" and "In line with current practice in the general population, our review confirms the substantial role of BMA in the diagnostic work up of FUO and Leishmania infection, also in KT recipients. Indeed, the procedure demonstrated high sensitivity and specificity, promptly detecting the parasite in most cases. Other diagnostic modalities reported in the literature were immunofluorescence-based serological tests for Leishmania antigens, ELISA for Leishmania-specific antibodies, and PCR analysis for Leishmania detection and typing on whole blood or tissue aspirate samples. Recipients with obvious cutaneous or mucocutaneous lesions could be reliably assessed by microscopic evaluation, histology, or culture of sample tissues".
Q10) Table 5 should include information regarding the treatment duration. This is important when one considers the risk of relapse.
A10) As requested, we added information regarding first-line (Table 5) and second-line (Table 6) treatments duration.
Reviewer 2 Report
The manuscript from Favi et al., brings an interesting review of the
Epidemiology, Clinical Characteristics, Diagnostic Workup, and Treatment Options of Leishmania Infection in Kidney Transplant Recipients. This manuscript relies on the use of comprehensively reviewed available literature.
The data are interesting, showing one part of the patients that have a high risk of losing their lives to leishmaniasis infection. Since they receive immunosuppressant drugs, the diagnostic and treatment are complicated.
There is valuable information that could help clinicians involved in the care of this complex group.
However, some minor issues which are listed below, have to be considered before a possible acceptance for publication in Tropical Medical and Infectious Disease:
1. The abstract has to have a maximum of 200 words, and the authors have 213 words. It needs to reduce a little.
2. Italics are missing through the text in specific words (Leishmania, Phlebotomus, Lutzomyia). These names need to go in italic.
Author Response
We are very grateful to Reviewer 2 for the positive feedbacks.
Q1) The abstract has to have a maximum of 200 words, and the authors have 213 words. It needs to reduce a little.
A1) As requested, the abstract has 199 words.
Q2) Italics are missing through the text in specific words (Leishmania, Phlebotomus, Lutzomyia). These names need to go in italic.
A2) We changed the writing for specific words, using Italic as approprite.